# Primary and Recall Immune Responses to SARS-CoV-2 in Breakthrough Infection

**DOI:** 10.3390/vaccines11111705

**Published:** 2023-11-09

**Authors:** Silvia D’Orso, Marta Pirronello, Alice Verdiani, Angelo Rossini, Gisella Guerrera, Mario Picozza, Manolo Sambucci, Andrea Misiti, Lorenzo De Marco, Antonino Salvia, Carlo Caltagirone, Emiliano Giardina, Luca Battistini, Giovanna Borsellino

**Affiliations:** 1Neuroimmunology Unit, Santa Lucia Foundation IRCCS, 00143 Rome, Italy; s.dorso@hsantalucia.it (S.D.); m.pirronello@hsantalucia.it (M.P.); a.verdiani@hsantalucia.it (A.V.); g.guerrera@hsantalucia.it (G.G.); m.picozza@hsantalucia.it (M.P.); m.sambucci@hsantalucia.it (M.S.); a.misiti@hsantalucia.it (A.M.); lorenzo.demarco@policlinicogemelli.it (L.D.M.); l.battistini@hsantalucia.it (L.B.); 2Medical Services, Santa Lucia Foundation IRCCS, 00179 Rome, Italy; a.rossini@hsantalucia.it (A.R.); a.salvia@hsantalucia.it (A.S.); 3Department of Clinical and Behavioral Neurology, Santa Lucia Foundation IRCCS, 00179 Rome, Italy; c.caltagirone@hsantalucia.it; 4Genomic Medicine Laboratory UILDM, Santa Lucia Foundation IRCCS, 00179 Rome, Italy; e.giardina@hsantalucia.it; 5Medical Genetics Laboratory, Department of Biomedicine and Prevention, Tor Vergata University, 00133 Rome, Italy

**Keywords:** SARS-CoV-2, vaccination, breakthrough infection, adaptive immune response

## Abstract

Breakthrough infections in SARS-CoV-2 vaccinated individuals are an ideal circumstance for the simultaneous exploration of both the vaccine-induced memory reaction to the spike (S) protein and the primary response to the membrane (M) and nucleocapsid (N) proteins generated by natural infection. We monitored 15 healthcare workers who had been vaccinated with two doses of Pfizer BioNTech BNT162b2 and were then later infected with the SARS-CoV-2 B.1.617.2. (Delta) variant, analysing the antiviral humoral and cellular immune responses. Natural infection determined an immediate and sharp rise in anti-RBD antibody titres and in the frequency of both S-specific antibody secreting cells (ASCs) and memory B lymphocytes. T cells responded promptly to infection by activating and expanding already at 2–5 days. S-specific memory and emerging M- and N-specific T cells both expressed high levels of activation markers and showed effector capacity with similar kinetics but with different magnitude. The results show that natural infection with SARS-CoV-2 in vaccinated individuals induces fully functional and rapidly expanding T and B lymphocytes in concert with the emergence of novel virus-specific T cells. This swift and punctual response also covers viral variants and captures a paradigmatic case of a healthy adaptive immune reaction to infection with a mutating virus.

## 1. Introduction

The severe acute respiratory syndrome coronavirus 2 (SARS-CoV-2) pandemic has resulted in almost 7 million deaths all around the globe and in 770 million confirmed cases as of September 2023 (https://www.who.int/emergencies/diseases/novel-coronavirus-2019, last accessed on 13 September 2023). The extraordinary efforts put into the development of vaccines and their subsequent distribution to the whole world have saved millions of lives from SARS-CoV-2 associated coronavirus disease 2019 (COVID-19) (https://doi.org/10.26099/whsf-fp90, last accessed on 13 September 2023) SARS-CoV-2 vaccines reduce the incidence and mortality of COVID-19 [1,2] by inducing a robust antibody production and by promoting the differentiation of functionally effective and persistent anti-virus immune memory cells [3]. Furthermore, all studies confirm that vaccine-induced protection from severe disease remains high even in breakthrough infections with different variants, including Alpha (B.1.1.7), Beta (B.1.351), Gamma (501Y.V3 or P1), Delta (B.1.617.2) and Omicron (BA.1, BA.2, BA.4, and BA.5) [4,5,6,7]. Nonetheless, time-dependent waning of antibody titres and of circulating virus-specific cells does occur, thus calling for the close investigation of immunological memory [8,9]. Moreover, repeated exposure to the same viral antigens by vaccination and natural infection may induce immunological exhaustion [10,11], whose signs would be carried by the virus-specific T cells of infected individuals also in the form of reduced reactivity to SARS-CoV-2. Breakthrough infections provide the intriguing opportunity to compare primary and recall responses in the same individual and to study in detail the nature and the kinetics of the variations in both the composition and the functional status of virus-specific adaptive immune cells. Here, we focused on the adaptive immune responses in a cohort of vaccinated healthcare workers who had received two doses of the Pfizer BioNTech BNT162b2 mRNA vaccine and were 6–8 months later infected with SARS-CoV-2 during the Delta wave. Blood from each subject was taken at different time points from the first positive swab (day 0) until negativisation with PCR. We evaluated anti-RBD antibody levels and the phenotype, functions and kinetics of spike (S)-, membrane (M)- and nucleocapsid (N)-specific T and B lymphocytes at different time points. We aimed at comparing the primary immune response induced by SARS-CoV-2 infection (M and N specific) to the recall immune response previously induced by vaccination (S specific). This knowledge may also inform vaccination strategies, particularly for the frail.

## 2. Materials and Methods

### 2.1. Human Subject Recruitment and Sampling

Fifteen health care workers at Santa Lucia Foundation were enrolled between July and October 2021 (Appendix A) during the Delta wave. All donors signed informed consent forms approved by the Ethical Committee of the Santa Lucia Foundation. These donors had received two doses of the Pfizer BioNTech BNT162b2 mRNA vaccine between December 2020 and January 2021. Venous blood was collected on the same day of the first PCR-based positive test and at different time points until negativisation. The donors were all mildly symptomatic and did not undergo treatment with steroids, monoclonal antibodies or antiviral drugs. Blood samples were collected in heparinized tubes and processed immediately. Absolute cell counts were obtained from each sample. In parallel, plasma samples were collected for the measurement of anti-receptor binding domain (RBD) antibodies.

### 2.2. Evaluation of Anti-SARS-CoV-2 Antibodies

The assessment of anti-RBD antibodies was conducted using an electrochemiluminescence sandwich immunoassay performed with the Roche Elecsys Anti-SARS-CoV-2 S kit (manufactured by Roche Diagnostics, Switzerland). Antibody levels were quantified using a Cobas 601 modular analyser, also from Roche Diagnostics, Switzerland, with a predefined threshold of 0.8 U/mL. The measurements in Elecsys Anti-SARS-CoV-2 S U/mL correspond to Binding Arbitrary Units per millilitre (BAU/mL). Before analysis, certain samples were diluted at a 1:4 ratio.

### 2.3. Antigen-Specific B Cell Detection

Peripheral blood mononuclear cells (PBMCs) were labelled with an 18-color flow cytometry panel comprising S- and RBD-fluorescent tetramers and antibodies against surface markers for the definition of different B cell subpopulations (Appendix A). Tetramers were synthesised by incubating SARS-CoV-2 S-Prot (HEK)-Biotin (Miltenyi, 25 μg) or SARS-CoV-2 RBD (HEK)-Biotin (Miltenyi, 50 μg) with fluorophore-conjugated streptavidin (streptavidin-PE (Miltenyi), streptavidin-PE Vio770 (Miltenyi) or streptavidin-iFluor810 (AAT Bioquest), respectively, at a molar ratio of 4:1 to generate three individual tetramers: S-PE, S-PE Vio770 and RBD-iFluor810. We used double discrimination to exclude B cells that non-specifically bound to fluorochrome or streptavidin moieties of the tetramers. Staining was performed as previously described [12].

### 2.4. Activation Induce Markers (AIM) Assay

PBMCs were isolated from freshly obtained whole blood with Ficoll–Paque density gradient centrifugation. Cells were seeded in U-bottom 96-well plates and stimulated for 18 h using PepTivator peptide pools (S, S+, S1, M, N) consisting of lyophilised 15 mer peptide sequences with 11 amino acids overlap, spanning the entire sequence of the S, M and N proteins of the SARS-CoV-2 Wuhan wild-type virus (all from Miltenyi, 1 μg/mL each). Purified αCD40 (0.5 μg/mL; Miltenyi) was added at the culture start.

### 2.5. Flow Cytometry Staining and Acquisition

PBMCs were collected and placed in V-bottom 96-well plates. They were subsequently resuspended in 30 μL of previously titrated antibodies (as listed in Appendix A) diluted in Brilliant Stain Buffer from BD Biosciences. The suspension was then incubated at room temperature (RT) for 15 min. The detailed information about the antibodies used for both surface and intracellular staining can be found in Appendix A. Following the staining and fixation process, the samples were promptly acquired using a fully equipped CytoFLEX LX within a 6 h timeframe. To ensure instrument performance, QC beads from Beckman Coulter were used daily for verification and calibration. Absolute cell counts were determined using a volumetric, lyse-no-wash, flow cytometry approach. For each sample, 50 μL of whole blood was stained with antibodies targeting CD19, CD3, CD4 and CD8 (as described in Appendix A). Next, 450 μL of the fluorescence-activated cell sorting lysing solution from BD Biosciences was added. After the lysis of red blood cells, the samples were processed on a CytoFLEX with a volume-calibrated peristaltic pump. The instrument automatically provided the counts of B cells, CD4+ T cells and CD8+ T cells per millilitre, which were then multiplied by 10,000 to determine the absolute counts in CD4+ and CD8+ T cells per millilitre. The absolute counts for S- and RBD-reactive B cells as well as AIM+, cytokine+ and related subsets were determined by multiplying their proportions by the absolute count of their parent lineages. The gating strategies employed for each staining can be found in Appendix A. Finally, the data were analysed using FlowJo version 10.8.1.

### 2.6. Intracellular Cytokine Staining (ICS) Assay

PBMCs were subjected to an hour-long incubation with either peptide pools or water, which served as a negative control. Subsequently, Monensin and Brefeldin A (both obtained from Sigma-Aldrich and used at concentrations of 5 μM and 10 μg/mL, respectively) were introduced into the cell cultures. After 17 h, the cells were initially stained for surface markers as previously described, followed by a thorough washing and fixation step in 3.6% formaldehyde at 4 °C for 5 min. Intracellular cytokine staining (ICS) was then carried out by incubating the cells in 30 µL of antibodies, which were appropriately diluted in a 0.5% *w*/*v* saponin solution from Sigma-Aldrich, as outlined in Appendix A.

### 2.7. Statistical Analysis

Statistical analyses were executed using Prism 10 (Graphpad Software, Boston, MA, USA). Specific information about the statistical tests employed for each experiment can be found in the corresponding figure. Time points were assessed through Kruskal–Wallis, Friedman, or Mann–Whitney tests, and post hoc Dunn’s tests were applied as necessary. Statistical significance was inferred when the *p*-value was less than 0.05.

## 3. Results

### 3.1. Kinetics of Circulating Anti-RBD Antibodies and Antigen-Specific B Cells during Breakthrough Infection in Subjects Vaccinated with Two Doses of SARS-CoV-2 mRNA Vaccine

To assess the immune response to SARS-CoV-2 breakthrough infection in vaccinated individuals, we measured anti-RBD antibody titres in serum samples collected on the day of the first positive COVID-19 test (day 0) and 2–5, 6–9 and 15–17 days later until negativisation (Figure 1a and Appendix A, for the study design). All individuals in our cohort had a consistent level of anti-RBD antibodies at baseline (median at day 0: 690.8 AU) as a result of the previous SARS-CoV-2 vaccination. Anti-RBD antibody production was quickly induced (days 2–5, median 18316 AU) and increased thereafter (days 6–9, median 31324 AU, and days 15–17, median 36592 AU). Thus, breakthrough infections rapidly and efficiently induced the production of high levels of RBD-specific antibodies. To characterise in detail the B cell response to SARS-CoV-2, we focused on antigen-specific B cells. Using three different fluorescent S and RBD tetramers [13,14] and a panel of fluorescent antibodies, we could single out virus-specific B cells (Appendix A). During breakthrough infection, we observed a significant increase in the absolute numbers of S-specific B cells at 6–9 days after infection, returning to baseline levels after 15–17 days (#cells/mL day 0: median 169; 2–5 days: median 187; 6–9 days: median 461; 15–17 days: median 232) (Figure 1b). Antigen-specific B cell frequencies expressed in % of CD19+ lymphocytes are displayed in Appendix A. Following the same trend, the absolute number of RBD-specific B cells increased significantly after 6–9 days with a peak at 15–17 days (#cells/mL day 0: median 9; 2–5 days: median 40; 6–9 days: median 108; 15–17 days: median 68). The frequency of RBD-specific B cells within S-specific B cells increased over time (day 0: median 5.33%; 2–5 days: median 14.26%; 6–9 days: median 33.46%; 15–17 days: median 42.42%) (Figure 1c). Altogether, we observed the rapid activation and expansion and the subsequent contraction of S-specific memory B cell responses in vaccinated individuals during breakthrough infection.

### 3.2. Deep Phenotypic Characterisation of S- and RBD-Specific B Cells

We then fully characterised the phenotype of circulating S- and RBD-specific (tetramer+) B cell subpopulations discriminating between seven distinct B cell subsets (Appendix A). During breakthrough infection, we did not observe significant fluctuations in the frequency of B cell subpopulations in the bulk B cell compartment. On the contrary, we observed a redistribution of B cell subpopulations in the S- and RBD-specific B cell pools (Figure 1d and Appendix A). As expected, the frequency of S- and RBD-specific antibody secreting cells (ASC) peaked at 2–5 days after infection and then returned to baseline after 15–17 days. The recall immune response in the antigen-specific memory B cell compartment also showed significant variations: classical memory switched (cmemSW) increased during infection, especially after 15–17 days, indicating persistent immunological memory. Also, these cells showed a trend towards increased levels of surface CD27 over time, suggesting affinity maturation through germinal centre reaction (Appendix A) [15]. The fraction of double negative 1 (DN1) cells remained constant in the S-specific B cell pool and decreased in the RBD-specific B cell compartment, while the frequency of the other subpopulations did not change during viral infection except for a significant increase in RBD-specific unswitched memory (USWmem) B cells after 15–17 days. B cells were also characterised in terms of IgM and IgD expression (Appendix A). As expected, the majority of S-specific and particularly of RBD-specific B cells were IgD- IgM- (switched, SW), and this fraction increased during infection.

We then focused on the subsets, which showed the most significant variations in frequency following infection, namely S-specific cmemSW, S-specific ASC and RBD-specific B cells. We measured the surface expression of CD11c, CD80, CD86, CD95, CXCR5 and CD73 (Figure 2 and Appendix A). CD11c increased significantly only in S-specific cmemSW and RBD-specific B cells, peaking after 15–17 days of infection, while as expected, this marker was not expressed on ASC. S-specific cmemSW and RBD-specific B cells expressed high levels of CD80, contrary to ASC S-specific B cells. CD95 and CD86 activation markers showed a comparable trend with low expression levels at baseline, peaking at 2 to 5 days and 6 to 9 days in all three subgroups examined and returning to baseline levels at 15 to 17 days. CXCR5, which directs cells to germinal centres in the lymph nodes, increased at 2–5 days and 6–9 days post-infection in S-specific ASC, while showing the opposite trend in RBD-specific B cells. In S-specific cmemSW, the CXCR5+ subset significantly decreased during infection, likely due to either the positioning of the CXCR5+ cells in the lymph nodes and/or generalised activation [16]. CD73 expression was not significantly modulated but showed a trend towards a reduction at the peak of the response.

### 3.3. Primary and Recall T Cell Immune Responses during Breakthrough Infection

We then assessed the antigen-specific T responses against S and non-S peptides (Appendix A for all gating strategies). First, we performed an activation induced marker (AIM) assay. All assays were performed on freshly isolated PBMCs, which were stimulated for 18 h with peptide pools spanning the entire sequence of S, M and N structural proteins. AIM+ CD4+T cells were defined by the upregulation of CD69+ and CD40L markers, whereas the upregulation of CD69+ and CD137+ defined AIM+ CD8+ T cells. After background subtraction from unstimulated control cultures, cell frequencies (Appendix A) and absolute cell counts were calculated at each time point. We observed that the antiviral T cell response at the time of the first positive PCR test was mainly due to the reactivation of S-specific CD4+ T cells; M-specific T cells were also detected, although at very low levels, while N-specific CD4+ T cell responses were almost absent (Figure 3a, left panel). After 2–5 days, a gradual response was observed for non-S peptides, which achieved the peak at 6–9 days. Moreover, during the contraction phase at 15–17 days, no significant differences were observed between T cell responses to S, M or N, although the fraction of S-specific T cells was slightly higher. Already at the first time point, the stimulation index (SI) of S-specific CD4+ T cells was >30 and increased to approximately 70 after two weeks, while M- and N-specific CD4+ cells responded less vigorously (Figure 3a, right panel). AIM+ CD8+ T cells were also detected with kinetics similar to those observed for CD4+ T cells, although with much lower magnitude (Figure 3b, left panel). Similar to CD4+ cell responses, CD8+ T cells also recognised peptides from the M and N proteins to smaller extents compared to S. S-specific T cells, detectable at very low levels on day 0, peaked at 6–9 days and then contracted. A small fraction of M-specific CD8+ T cells was already present at day 0, probably due to previous exposure and cross-reactivity to other SARS-CoV strains. Again, the S-specific CD8+response was predominant at all time points, also in terms of SI (Figure 3b, right panel).

### 3.4. Cytokine Production during Primary and Recall T Cell Responses

To fully capture the features of the virus-specific T cell response, we also performed intracellular cytokine staining (ICS) after overnight exposure to peptide pools and assessed cytokine production by antigen-specific T cells. At day 0, we detected a substantial fraction of memory S-specific IFN-γ+ CD4+ and CD8+ T cells together with a small fraction of M-specific IFN-γ+ CD4+ and CD8+ T cells, while those specific for the N protein were virtually absent (Figure 3c,d, left panels). At the second measurement, S-specific IFN-γ+ CD4+ and CD8+ T cells were greatly increased, and M-specific IFN-γ+ CD4+ T cells were now present in significant numbers, while sparse N-specific IFN-γ+ CD4+ T cells appeared only at the 6–9 days window. During the contraction phase (15–17 days), IFN-γ+ responses to the three peptide pools were not significantly different, probably due to the small number of samples. As in the AIM assay, the S-specific T cell population was predominant, and the SI of both S-specific IFN-γ+ CD4+ and CD8+ was higher than that of M- and N-specific IFN-γ+ CD4+ and CD8+ T cells at all time points (Figure 3c,d, right panels, and Appendix A for cell frequencies). The kinetics of CD4+ and CD8+ T cell responses are shown separately for each peptide pool (Appendix A). Thus, breakthrough infections with the Delta variant in vaccinated individuals elicited a substantial T cell response against the vaccine immunogen (S protein) together with a slightly delayed emergence of T cells specific for other viral components, particularly the M protein. The poor CD8+ T cell response, particularly to the M and N proteins, may be due to the peptides used in the AIM and ICS assays, which stimulate CD4+ cells more efficiently than CD8+ cells [17].

### 3.5. Deep Characterisation of the Phenotype and Functionality of Antigen-Specific CD4+ T Cells

As our donors were all vaccinated at the time of infection with SARS-CoV-2, we had the intriguing opportunity to compare the features of recall and primary immune responses. Thus, we deepened the characterisation of S- (vaccine-induced), M- and N-specific (first encounter) CD4+ T cells at all time points only in subjects with clearly detectable CD4+ T cell responses. First, we assessed the differentiation status of both AIM+ and total CD4+ cells for each peptide stimulation and time point through the detection of CD45RA and CCR7 expression (Figure 4a,b). In total CD4+ cells, the naïve compartment was the largest at all time points, followed by effector memory (EM) and central memory (CM), with terminally differentiated effector memory (EMRA) cells being the less represented, as already described [12]. In contrast, the S- and M-specific AIM+ CD4+ at day 0 comprised mostly CM and EM subsets and a small fraction of naïve and EMRA cells. The fraction of naïve cells significantly decreased during infection, while EM became the predominant subset. Although not reaching statistical significance, N-specific AIM+ CD4+ cells seemed to have the same tendency as S- and M-specific cells. Notably, no differences between the S-specific and M- or N-specific CD4+ cells were detected in the distribution of the differentiation subsets at any time point. Then, we evaluated the expression of activation/exhaustion and homing cell markers (CD25, CD38, HLA-DR, CXCR5, ICOS and PD-1) (Figure 5a). At day 0, S-specific CD4+ cells were already activated and showed higher levels of CD25+, CXCR5+, ICOS+ and PD-1+ compared to M-specific cells. Unexpectedly, at the first time point, N-specific CD4+ cells seemed to have an activated phenotype. During the course of infection, activation markers further increased in S-specific CD4+ lymphocytes with a peak at approximately 2 to 5 days from the first positive swab and then slightly raised (ICOS+), rapidly decreased (CD25+, CD38+ and HLA-DR+ and CXCR5+) or remained stable (PD-1+) until the last time point. The activation markers in non-S-specific CD4+ cells followed the same trend as their S-specific counterpart. Furthermore, we deepened the evaluation of the ability of S-, M- and N-specific CD4+ cells to produce cytokines (Figure 5b). In S-specific CD4+ T cells, both the numbers of IL-2+ and polyfunctional IFN-γ/IL-2 double-positive cells peaked at approximately 6 to 9 days and were higher than in M- and N-specific CD4+ T cells. IL-2+ N-specific CD4+ appeared to have a similar tendency, although the small number of responding subjects might mask a different kinetic. Finally, we searched for the stable population of memory cells normally established following infection: T stem cell memory (TSCM) cells, a rare subset of lymphocytes able to self-renew and replenish the pool of memory T cells following in vivo rechallenge [18]. At day 0, the number of S-specific TSCM CD4+ cells was significantly higher than M- and N-specific TSCM CD4+. S-specific TSCM CD4+ cells sharply increased in number during the infection, while M- and N-specific TSCM CD4+ cells showed a slower and less prominent increase. Interestingly, the numbers of TSCM CD4+ specific for S, M and N were comparable in the last time point, suggesting the establishment of a memory reservoir also for the novel viral antigens from the M and N proteins (Figure 5c). To summarise, S-specific secondary immune response showed a faster and more pronounced increase in activation markers and cytokine production compared to the M- and N-specific primary immune response. However, at the later time points, M- and N-specific cells had the same features of S-specific T cells with whom they also shared the kinetics.

## 4. Discussion

Breakthrough infection with SARS-CoV-2 and its variants has been shown to induce rapid recovery of memory and de novo adaptive immune cell responses [6,19,20,21,22,23]. Although our cohort was small with only 15 participants, we could monitor the hold of the S-specific B and T cell recall responses induced by vaccination with Pfizer BioNTech’s BNT162b2 during the SARS-CoV-2 Delta (B.1.617.2) wave and follow the primary T cell response directed against the M and N proteins of SARS-CoV-2. During infection, anti-RBD neutralising antibodies quickly rose to high levels [13,14]. Using S and RBD tetramers, we could precisely identify and characterise S-specific B cells with unprecedented detail. Vaccine-derived memory B lymphocytes rapidly increased in number, and S-specific ASCs expanded and showed signs of activation already at 2–5 days post infection [24,25].

Phenotypic analysis of B cells showed an increase in the fraction of the cmemSW subset, which represented 6% (range: 2.44–23.48%) of the total B cells and were 5-fold enriched in the vaccine-induced S-specific subsets. Importantly, at the last time point, the cmemSW subset was the most represented within S-specific B cells and showed an activated phenotype, a progressive loss of CXCR5 expression, likely accounting for their recent release from lymphoid follicles, and a steady increase in CD11c. In addition, as a surrogate marker of affinity maturation having occurred in the germinal centre, S-specific-cmemSW B cells increased the expression levels of CD27 [15,26], further suggesting that successful germinal centres had been established.

DN B cells are thought to derive from the extrafollicular pathway and do not undergo somatic hypermutation; they are divided into DN1, defined as low affinity [27], and DN2, enriched in autoreactive cells related to chronic inflammation [28]. Several reports have shown an increase in the DN2 subset during the course of severe COVID-19 in unvaccinated people [29,30,31]. In contrast, DN B cells have been reported to decrease [32] in the setting of mRNA vaccination. Here, we analysed not only the total B population but also the virus-specific subsets, and in our setting of mild COVID-19 in vaccinated individuals, we found that the DN2 subset remained substantially stable, while the DN1 subset decreased. This decrease in DN1 cells with low affinity receptors may reflect a refinement of the immune response and the successful generation of high-affinity antibodies by germinal centre-derived B cells. In the context of T-independent memory B cell generation, we detected an increase in USWmem within the RBD-specific B cell compartment; as previously demonstrated, this population correlates positively with anti-RBD antibody titres and negatively with duration of symptoms [33]. Thus, the body of information gained by our careful investigation on the phenotype of the S- and RBD-specific B cell pool may, at least in part, explain the rapid protection conferred by a vaccine-induced/infection-evoked antigen-specific B cell response in the otherwise highly virulent SARS-CoV-2 Delta variant.

This S-specific B cell activation was paralleled by the concomitant activation of SARS-CoV-2-specific T cell responses. Recall T cell response to the S protein was characterised by a rapid expansion of antigen-specific memory T cells with SI > 30 and expressing high levels of activation markers in addition to markers of functional differentiation. S-specific T cells were mostly EM and CM with a very minor fraction presenting the terminally differentiated EMRA phenotype. Also, S-specific T cells demonstrated a strong effector capacity (such as IFN-γ and IL-2 production). Early after infection, we also observed the presence of polyfunctional memory CD4+ T cells capable of producing, simultaneously, IL-2 and IFN-γ. The T cell subset expressing high levels of CXCR5, PD-1 and ICOS is specialised in aiding B cell affinity maturation and antibody production. These T follicular helper cells increased jointly with B cell activation, as already reported [34,35]. We also observed S-specific cells with a TSCM phenotype. These cells were clearly detectable at the time of infection and represent the long-lived pool of virus-specific memory cells that survive the contraction phase of the immune response [12,36].

M- and N-specific T cells showed the features of a blossoming primary response following kinetics akin to S-specific T cells but of a smaller magnitude. Responses to the M protein were detectable already at the first time point, whilst N-specific T cells were barely detectable. This may be explained by cross-reactivity of M-specific T cells due to T cell receptor (TCR) degeneracy, which allows for the recognition of several similar peptides [37]. For instance, it has been shown that Common Cold Coronaviruses-specific T cells cross-reactive with SARS-CoV-2 are present in most individuals, and it has been proposed that these cells may enhance anti SARS-CoV-2 immune responses [38,39].

These data provide further evidence that exposure of vaccinated individuals to viral antigens during natural infection with SARS-CoV-2 variants induces a paradigmatic immune response carried out by both a legion of vaccine-induced, antigen-experienced memory B and T cells and by newly emerging virus-specific cells.

## 5. Conclusions

Our study demonstrates that SARS-CoV-2 breakthrough infections and its variants trigger rapid recovery of memory adaptive immune cell responses. In our cohort, we monitored the persistence of S-specific B and T cell recall responses in BNT162b2-vaccinated individuals experiencing breakthrough infections during the SARS-CoV-2 Delta wave. Infection led to a rapid increase in anti-RBD neutralising antibodies, and a detailed analysis of S-specific B cells revealed a surge in vaccine-derived memory B lymphocytes and activated S-specific antibody-secreting cells. Additionally, SARS-CoV-2-specific T cell responses, particularly to the S protein, showed a robust and diverse effector capacity, and the presence of polyfunctional memory CD4+ T cells. These findings shed light on the robust and multifaceted immune response in vaccinated individuals facing SARS-CoV-2 variants during natural infection.

## Figures and Tables

**Figure 1 vaccines-11-01705-f001:**
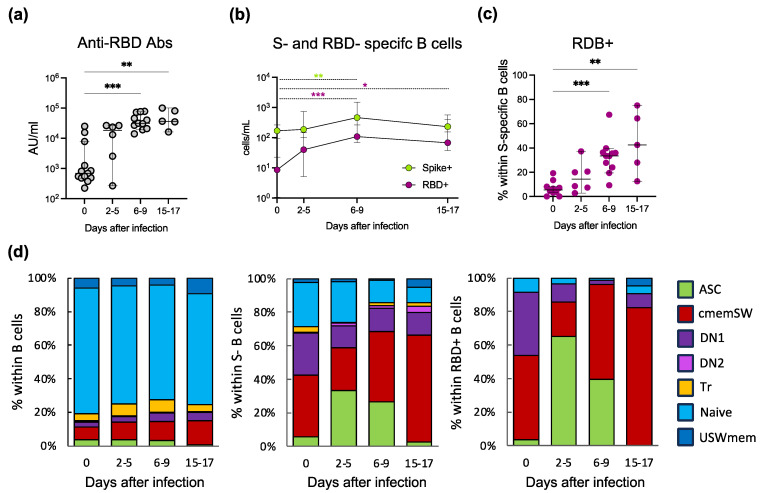
Antibody and B cell responses during breakthrough infection. (**a**) Serum anti-RBD antibody dosage (day 0, *n* = 12; 2–5 days, *n* = 6; 6–9 days, *n* = 11; 15–17 days, *n* = 5). (**b**) Absolute cell counts of S- and RBD-specific B cells (day 0, *n* = 12; 2–5 days, *n* = 6; 6–9 days, *n* = 11; 15–17 days, *n* = 5). Statistical significance of the comparisons is indicated by asterisks coloured according to S- or RBD-specificity. (**c**) Frequency of RDB-specific B cells within S-specific B cells (day 0, *n* = 6; 2–5 days, *n* = 6; 6–9 days, *n* = 11; 15–17 days, *n* = 5). Values were compared with nonparametric repeated measures Kruskall–Wallis and corrected for Dunn’s multiple comparison tests; * *p* < 0.05; ** *p* < 0.01; *** *p* < 0.001; (**d**) Frequency of the different B cell subpopulations within total B-, S- and RBD-specific cells. (RBD: Receptor Binding Domain, S: Spike, ASC: Antibody Secreting Cells, cmemSW: classical memory switched, DN1: double negative 1, DN2: double negative 2, Tr: transitional, USWmem: unswitched memory).

**Figure 2 vaccines-11-01705-f002:**
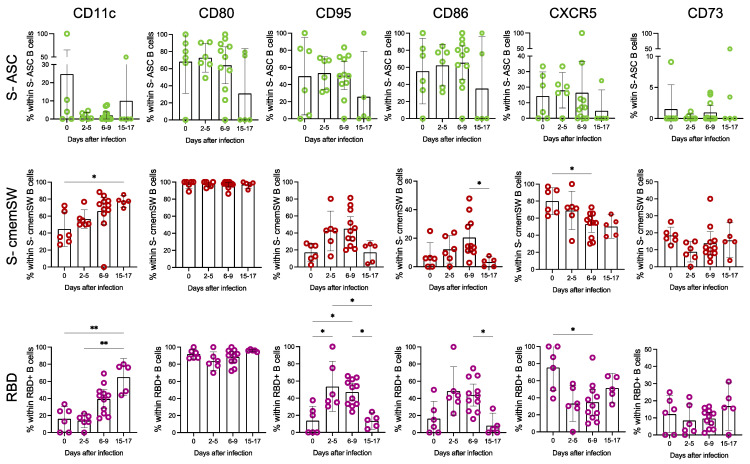
Phenotypic characterisation of B cells during breakthrough infections. Frequency of CD11c, CD80, CD95, CD86, CXCR5 and CD73 within S-specific ASC (green circles), S-specific cmemSW (red circles) and RBD-specific B cells (purple circles) (day 0, *n* = 6; 2–5 days, *n* = 6; 6–9 days, *n* = 11; 15–17 days, *n* = 5). Values were compared with nonparametric repeated measures Kruskall–Wallis and corrected for Dunn’s multiple comparison tests; * *p*< 0.05; ** *p*< 0.01. (S: Spike, RBD: Receptor Binding Domain, ASC: Antibody Secreting Cells, cmemSW: classical memory switched).

**Figure 3 vaccines-11-01705-f003:**
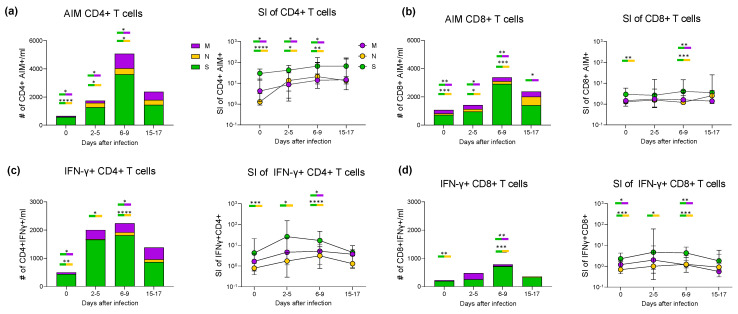
CD4+ and CD8+ T cell responses during breakthrough infection. (**a**,**b**) In the left panels, the graphs illustrate the median absolute cell counts of CD4+ (**a**) or CD8+ (**b**) T cells specific for S, N and M antigens as determined with the AIM assays performed at the indicated time points. The right panels display the SI for CD4+ and CD8+ antigen-specific cells. (**c**,**d**) The graphs in the left panels show absolute cell counts of IFN-γ-producing CD4+ (**c**) and CD8+ (**d**) T cells, and right panels show the SI. Non-parametric Friedman tests followed by Dunn’s post hoc tests were used to compare responses to S and non-S peptides from the same donor at the same time points. Median values obtained at each time point for S-, M- and N-specific T cells were compared with non-parametric Friedman tests followed by Dunn’s post hoc tests. Statistical significance of the comparisons is indicated by asterisks positioned over the rods coloured according to S, M or N reactivity. * *p* < 0.05; ** *p*< 0.01; *** *p* < 0.001; **** *p* < 0.0001; no symbol, not significant. (S: Spike, M: Membrane, N: Nucleocapsid, AIM: Activation Induced Markers).

**Figure 4 vaccines-11-01705-f004:**
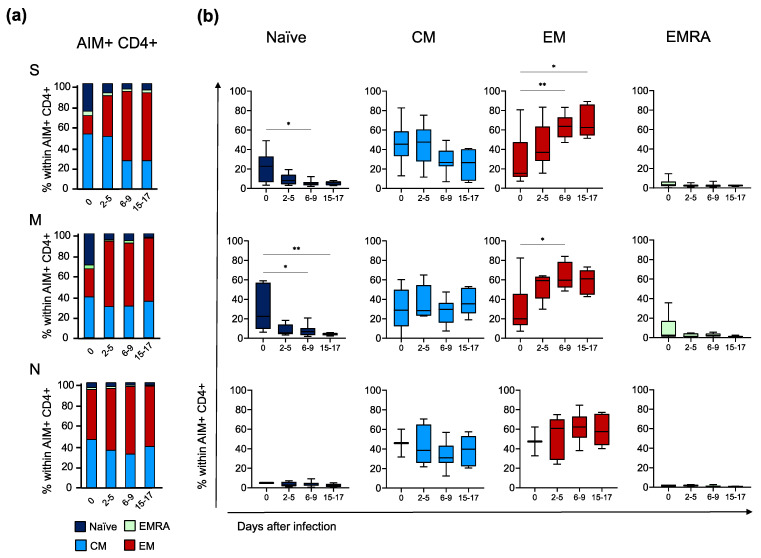
Differentiation status of virus-specific CD4+ T cells during breakthrough infection. (**a**) Fractions of naïve, terminally differentiated effector memory (EMRA), effector memory (EM) and central memory (CM) within AIM+ CD4+ T cells compared with time points for each peptide S (top), M (middle) or N (bottom). (**b**) Frequency of naïve, EMRA, EM and CM within AIM+ CD4+ T cells compared with time points for S (top), M (middle) or N (bottom). Time points were compared with nonparametric repeated measures Krukall–Wallis and corrected for Dunn’s multiple comparison tests; lines represent median with 5 to 95th percentile. * *p* < 0.05; ** *p* < 0.01; no symbol, not significant. (S: Spike, M: Membrane, N: Nucleocapsid, AIM: Activation Induced Markers).

**Figure 5 vaccines-11-01705-f005:**
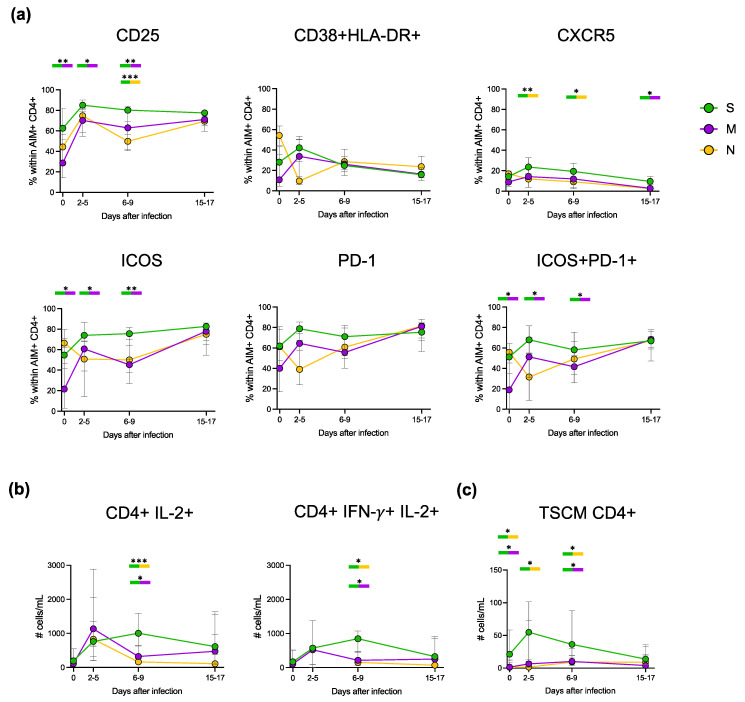
Phenotype and functionality of antigen-specific CD4+ T cells in responder subjects. (**a**) Frequency of different activation markers within AIM+ CD4+ T cells compared with time points for each peptide: S, M and N. Time points within each peptide were compared with nonparametric Mann–Whitney; bold lines represent the median of plotted values. * *p* < 0.05; ** *p* < 0.01; *** *p* < 0.001; no symbol, not significant. (**b**) Longitudinal analysis of absolute cell counts showing IL-2 alone (left), and IFN-γ+ and IL-2+ (right) production within CD4+ T cells for each peptide. Statistical analysis was performed as in A. (**c**) Longitudinal analysis of absolute cell counts of TSCM within CD4+ T cells for each peptide. Statistical analysis was performed as in (**a**). (S: Spike, M: Membrane, N: Nucleocapsid, AIM: Activation Induced Markers, TSCM: T Stem Cell Memory).

## Data Availability

The datasets used and/or analysed during the current study are available from the corresponding author on reasonable request.

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
