# Peer review of "Primary and Recall Immune Responses to SARS-CoV-2 in Breakthrough Infection"

_vaccines, 2023, doi:10.3390/vaccines11111705_

Round 1

Reviewer 1 Report

Comments and Suggestions for Authors

In the present study D’Orso and coworkers describe the activation of B and T cell responses at short time frame after SARS-CoV-2 infection. The study subjects include (healthy) adults that had been vaccinated two times with Pfizer/BioNtech mRNA vaccine ca. 6 months prior to contracting a breakthrough infection. The number of individuals in the study is 15 and a variable number of plasma and blood cell samples is available during the acute and convalescent phase of the disease. The authors have carried out a detailed analysis of B and T cell responses by staining the cells with an extensive number of cellular markers and thus the amount of data is high. The authors show that spike-specific B and T cell responses are rapidly activated during the infection while the responses to M and N protein peptides is only moderate or weak, respectively. The study includes a lot of information even though the number of study subjects is limited.

1.       It would have been very interesting to include study subjects that had only been vaccinated without a breakthrough infection and compare antibody and B and T cell responses to those individuals who were infected after vaccination. This approach would have provided more information on the long-term immunity in response to vaccination only.

2.       In Figure 1 antibody responses to the spike or RBD proteins have been described. It would also be very interesting to see whether there were any detectable antibodies against M and N proteins. This analysis would give basic information on the kinetics of the activation of primary immune responses (anti-M and anti-N) and compare those to recall responses (anti-S).

3.       All figure legends should include the description of the abbreviations so that the reader would not have to look at the results section for abbreviatons. This demand covers all figure and supplementary figure legends.

4.       It should be mentioned/described in the figures or in the text that what is the actual percentage or proportion of SARS-CoV-2 specific B and T cells among the whole population. In supplementary figure 2 spike-specific B cells include ca. 0.6% of the whole population. This information would be very interesting throughout the study i.e. is the percentage of antigen-specific B and T cells in the range of ca. 0.01% to 1%?

5.       Supplementary figures S2 and S7 are the same. Thus, S7 should include the data on T cells and not B cells.

6.       In references the authors should check the situation of one of the references which is referred as bioRxiv (17). Is the study already published or is there an alternative reference? Formally, bioRxiv refers to unpublished data. This is an important topic, since the paper refers to a possible exhaustion of T cell responses after repeated antigen exposures. At the moment to my knowledge such a phenomenon has so far not been described in COVID vaccinated individuals with or without breakthrough infection even if up 3-4 vaccine doses have been given. This is an important topic, but the statements have to be based on real published data.  

7.       Otherwise the paper is well-written and data extensive even though the number of study subjects is low. This fact could be mentioned in the discussion as one of the drawbacks of the study.

Author Response

Reviewer 1

In the present study D’Orso and coworkers describe the activation of B and T cell responses at short time frame after SARS-CoV-2 infection. The study subjects include (healthy) adults that had been vaccinated two times with Pfizer/BioNtech mRNA vaccine ca. 6 months prior to contracting a breakthrough infection. The number of individuals in the study is 15 and a variable number of plasma and blood cell samples is available during the acute and convalescent phase of the disease. The authors have carried out a detailed analysis of B and T cell responses by staining the cells with an extensive number of cellular markers and thus the amount of data is high. The authors show that spike-specific B and T cell responses are rapidly activated during the infection while the responses to M and N protein peptides is only moderate or weak, respectively. The study includes a lot of information even though the number of study subjects is limited.

  1. It would have been very interesting to include study subjects that had only been vaccinated without a breakthrough infection and compare antibody and B and T cell responses to those individuals who were infected after vaccination. This approach would have provided more information on the long-term immunity in response to vaccination only.

We agree that this is an interesting point. However, the long-term immunity provided by the vaccine can at least in part be evaluated by measuring T and B cell responses on the day of the positive swab (Day 0), since testing was performed immediately after the first symptoms and adaptive immune cells will not have had the time to expand. The donors in our cohort had all been vaccinated 6-8 months prior to the infection, and all showed a significant degree of immunological memory to the Spike protein, both humoral (Median Anti-S IgG on Day 0: 690 AU) and cell-mediated (Frequency of AIM+ CD4 cells: 0,12% of total CD4 cells; Frequency of AIM+ CD8 cells: 0,36% of total CD8 cells).

  1. In Figure 1 antibody responses to the spike or RBD proteins have been described. It would also be very interesting to see whether there were any detectable antibodies against M and N proteins. This analysis would give basic information on the kinetics of the activation of primary immune responses (anti-M and anti-N) and compare those to recall responses (anti-S).

We agree that this would have been a very interesting parallel to make, but at the time the study was performed the anti M and N reagents were not available, and we no longer have the serum samples.

  1. All figure legends should include the description of the abbreviations so that the reader would not have to look at the results section for abbreviatons. This demand covers all figure and supplementary figure legends

Thank you, we have corrected this.

  1. It should be mentioned/described in the figures or in the text that what is the actual percentage or proportion of SARS-CoV-2 specific B and T cells among the whole population. In supplementary figure 2 spike-specific B cells include ca. 0.6% of the whole population. This information would be very interesting throughout the study i.e. is the percentage of antigen-specific B and T cells in the range of ca. 0.01% to 1%?

We have added the data on antigen-specific B and T cells, which we show in two new Supplementary Figures (Fig.S3 and Fig. S9).

  1. Supplementary figures S2 and S7 are the same. Thus, S7 should include the data on T cells and not B cells.

We apologize for the error. We have now corrected Fig.S7 (now Fig. S7) which shows the gating strategy for T cells.

  1. In references the authors should check the situation of one of the references which is referred as bioRxiv (17). Is the study already published or is there an alternative reference? Formally, bioRxiv refers to unpublished data. This is an important topic, since the paper refers to a possible exhaustion of T cell responses after repeated antigen exposures. At the moment to my knowledge such a phenomenon has so far not been described in COVID vaccinated individuals with or without breakthrough infection even if up 3-4 vaccine doses have been given. This is an important topic, but the statements have to be based on real published data.  

We thank the reviewer for pointing this out. The paper of reference 17 is now published in Nature Immunology, and the authors show that prior vaccination actually enhances the activation and expansion of spike-specific cells, mostly of the central memory subset, without alterations of the broader T cell landscape and with no impairment of virus-specific T cell functionality in vaccinated individuals with breakthrough infections. We have updated the reference.

  1. Otherwise the paper is well-written and data extensive even though the number of study subjects is low. This fact could be mentioned in the discussion as one of the drawbacks of the study.

Thank you for this positive comment. We have included the statement that our sample size was low.

Reviewer 2 Report

Comments and Suggestions for Authors

Estimated Authors of the paper "Primary and recall immune responses to SARS-CoV-2 in breakthrough infection",

I've read with great interest the present research on the characteristics of immune response in SARS-CoV-2 breakthrough infection from a small but quite well characterized set of 15 HCWs who were initially vaccinated between 2020-2021.

According to the Authors, "natural infection with SARS-CoV-2 in vaccinated individuals induces fully functional and rapidly expanding T and B lymphocytes", that was characterized in several sub-sets of immune cells. MOre interestingly, Authors identified the the emergence of virus-specific T cells strains and clonal expansion from both T and B cell was able to cover, at least as suggested by laboratory analyses, viral variants. In other words, the present study suggests that vaccination is able to elicit a basic activation of the immune system that, in case of further encounters with new variants of the pathogen, will guarantee some degree of protection.

For all of the aforementioned reasons, the present paper - whose content is clearly written, and whose design is simple and correct, could be accepted without further adjustments, at least from my point of view.

Author Response

Reviewer 2

Estimated Authors of the paper "Primary and recall immune responses to SARS-CoV-2 in breakthrough infection",

I've read with great interest the present research on the characteristics of immune response in SARS-CoV-2 breakthrough infection from a small but quite well characterized set of 15 HCWs who were initially vaccinated between 2020-2021.

According to the Authors, "natural infection with SARS-CoV-2 in vaccinated individuals induces fully functional and rapidly expanding T and B lymphocytes", that was characterized in several sub-sets of immune cells. MOre interestingly, Authors identified the the emergence of virus-specific T cells strains and clonal expansion from both T and B cell was able to cover, at least as suggested by laboratory analyses, viral variants. In other words, the present study suggests that vaccination is able to elicit a basic activation of the immune system that, in case of further encounters with new variants of the pathogen, will guarantee some degree of protection.

For all of the aforementioned reasons, the present paper - whose content is clearly written, and whose design is simple and correct, could be accepted without further adjustments, at least from my point of view.

We thank the Reviewer for the positive comments.

Reviewer 3 Report

Comments and Suggestions for Authors

D'Orso et al study Breakthrough infections in individuals vaccinated with SARS-CoV-2 exploring both the memory reaction induced by the vaccine to the Spike protein (S) and the primary response to the membrane proteins (M) and Nucleocapsid (N) generated by natural infection. The authors show data demonstrating that exposure of vaccinated individuals to viral antigens during natural infection with SARS-CoV-2 variants induces a paradigmatic immune response carried out by both a legion of vaccine-induced antigen-experienced memory B and T cells and by newly emerging virus-specific cells. 

This interesting article is well written and well structured. The figures are clear and the references are appropriate and current. In the supplementary material I really like Figure S1. In my opinion it can be accepted in its current form.

Author Response

Reviewer 3

D'Orso et al study Breakthrough infections in individuals vaccinated with SARS-CoV-2 exploring both the memory reaction induced by the vaccine to the Spike protein (S) and the primary response to the membrane proteins (M) and Nucleocapsid (N) generated by natural infection. The authors show data demonstrating that exposure of vaccinated individuals to viral antigens during natural infection with SARS-CoV-2 variants induces a paradigmatic immune response carried out by both a legion of vaccine-induced antigen-experienced memory B and T cells and by newly emerging virus-specific cells. 

This interesting article is well written and well structured. The figures are clear and the references are appropriate and current. In the supplementary material I really like Figure S1. In my opinion it can be accepted in its current form.

We thank the Reviewer for the positive comments.
